# Innovative Approaches in Bone Tissue Engineering: Strategies for Cancer Treatment and Recovery

**DOI:** 10.3390/ijms26093937

**Published:** 2025-04-22

**Authors:** Samiksha S. Khobragade, Manish Deshmukh, Ujwal Vyas, Rahul G. Ingle

**Affiliations:** Datta Meghe College of Pharmacy, Datta Meghe Institute of Higher Education and Research, Sawangi (M), Wardha 442001, India

**Keywords:** artificial intelligence, bone tissue engineering, 3D printing, gene therapy, metastasis, scaffold fabrication, stem cells

## Abstract

Cancer has rapidly emerged as a leading global cause of premature mortality, with significant economic implications projected to reach USD 25.2 trillion from 2020 to 2050. Among the various types of cancer, primary bone cancers, though uncommon, are projected to see nearly 4000 new cases diagnosed in the United States in 2024. The complexity of treating bone cancer arises from its rarity, diversity, and the challenges associated with surgical interventions, metastatic spread, and post-operative complications. Advancements in bone tissue engineering (BTE) have introduced innovative therapeutic approaches to promote bone regeneration and address tumor recurrence. This interdisciplinary field integrates biomaterials, scaffolds, and gene therapy, utilizing technologies such as 3D bioprinting to create custom scaffolds that facilitate cellular activities essential for tissue regeneration. Recent developments in biodegradable, bioactive materials aim to enhance the biocompatibility and effectiveness of scaffolds, while nanotechnology presents promising avenues for targeted drug delivery and improved therapeutic outcomes. This review outlines the current landscape of BTE, highlighting scaffold fabrication techniques, the advantages of incorporating stem cell and gene therapies, and future directions, including the integration of artificial intelligence in scaffold design for personalized medicine in orthopedic oncology. This work underscores the necessity for ongoing research and innovation, aiming to improve therapeutic strategies specifically designed to address the unique challenges posed by bone sarcomas and metastatic cancers.

## 1. Introduction

Cancer is emerging as a leading cause of premature death worldwide [1]. The significant worldwide economic burden of cancer from 2020 to 2050 is anticipated to reach USD 25.2 trillion [2]. The American Cancer Society estimates that there will be approximately 3970 new cases of primary cancer of the bones and joints in 2024, with 2270 cases in males and 1700 in females. It also estimates about 2050 deaths from this cancer (1100 in males and 950 in females). Osteosarcoma is the most common type of primary bone cancer, followed by chondrosarcoma and Ewing sarcomas [3].

Bone tumors, which invade bone tissue, can be categorized as primary or metastatic [4]. The major treatment method for bone cancer is surgical excision of malignancies. Pre-operative therapies, especially post-operative adjuvant anticancer methods, are critical for achieving excellent therapeutic outcomes. However, the occurrence of post-operative bone tumor recurrence, metastasis, substantial bone abnormalities, and infection are important hazards that might result in poor prognoses or even treatment failure. The treatment of bone cancer presents numerous problems, including the disease’s rarity and diversity, limitations in effective systemic medicines, difficulties with pain management, and the complex nature of surgical procedures. Addressing these problems requires constant research and innovation to develop more effective treatment techniques suited to the specific characteristics of bone sarcomas [5]. In recent years, substantial progress has been made in the development of tissue engineering (TE) in the area of bone cancer therapy and bone regeneration [6]. The purpose of this review is to explore and develop innovative approaches in bone tissue engineering (BTE) that specifically address the unique challenges faced in cancer treatment and recovery. Patients with cancer frequently experience complications related to bone integrity, including metastasis, lesions, and treatment-related skeletal side effects. This article aims to bridge the gap between oncology and regenerative medicine by enhanced bone regeneration, targeted drug delivery, multifunctional scaffolds, and functional integration.

BTE is an interdisciplinary field that uses biomaterials, scaffolds, cells, and regulatory signals to create site-specific structural and biological similarities during metastasis. These three pillars of BTE proposed by Langer and Vacanti [7] are shown in Figure 1. Advances in BTE have resulted in creating porous microstructure scaffolds that can be seeded with human cells and growth factors for in vitro cancer metastasis studies. TE can revolutionize cancer research by enabling direct observation of tumorigenesis and migration at metastasis, a hallmark of cancer, which often involves neovascularization [8,9]. TE provides a viable option for bone regeneration by eliminating the requirement for tissue donors and allowing for a single surgical intervention, potentially lowering implant failure rates [10,11]. Recent advancements in TE include gene-editing technologies such as CRISPR/Cas9 to produce scaffolds that aid in tissue healing and regeneration. TE prompted the creation of novel approaches for regenerative medicine and disease modelling. Among the most significant recent advances in TE include the utilization of 3D bioprinting, organ-on-a-chip, and induced pluripotent stem cell technologies.

Primarily, BTE focuses on developing biocompatible, osteoconductive, or osteoinductive biomaterials for tissue regeneration [12]. Unfortunately, many of them showed a lack of cytocompatibility. According to the literature, poly (lactide-co-glycolide) (PLGA) biomaterials release acidic byproducts during degradation, which could cause serious tissue necrosis and implant failure. On the other hand, polyphosphazenes produce neutral or basic compounds during the degradation process. To address the issue of acidic byproducts from poly(lactide-co-glycolide) (PLGA) in BTE, incorporating basic or bioactive fillers such as hydroxyapatite, calcium carbonate, or magnesium hydroxide into the PLGA scaffold can help neutralize the acidic environment while enhancing osteoconductivity. Additionally, blending PLGA with polyphosphazene, also releasing neutral or basic degradation products, creates a hybrid material with balanced pH and improved biocompatibility. Surface modifications, such as coating PLGA with bioactive glass or grafting alkaline functional groups, can further mitigate acidity. Employing slow-release alkaline buffers or adopting alternative polymers like polycaprolactone (PCL) can also ensure a stable environment conducive to bone regeneration, minimizing tissue damage and implant failure.

The current trend in biomaterials has represented biomimetic biomaterials (comprising organic and inorganic components) that would have the ability to mimic a natural bone environment, accelerating the regeneration process [13]. Tissue-engineered scaffolds offer physical support for bone defects and multimodal treatment of bone cancer, with nanoparticulate delivery systems repressing tumor growth and aiding new bone formation [14]. Nanoparticles (NPs) are at the forefront of nanotechnology, and their unique size-dependent features have shown promising impact on challenges faced by TE.

To date, the standard treatment for bone cancer combines surgery with chemotherapy, but it is laden with complications such as post-operative tumor recurrence and chemotherapy-induced adverse effects. BTE offers a revolutionary method for bone defect healing by using bioactive scaffolds made of metals, ceramics, and hydrogels. BTE has made outstanding progress in correcting skeletal abnormalities, especially in the context of tumoral pathologies. It provides promising ways for rebuilding injured bone. However, its implementation in the setting of tumoral pathologies such as osteosarcoma and bone metastases presents unique challenges. This includes removing cancerous cells, bone deformities, scaffolds, biocompatibility, biodegradability, vascularization, and patient-specific aspects. These limitations can be addressed by developing a multifunctional treatment platform that includes chemotherapeutic medications, photothermal agents (PTAs), photosensitizers (PSs), sound sensitizers (SSs), magnetic thermotherapeutic agents (MTAs), and naturally occurring anticancer chemicals [15].

Tumor-altered bone microenvironments have disturbed signaling pathways, reducing the efficacy of engineered constructs. Recent advances have included bioactive scaffolds like hydroxyapatite-functionalized gold NPs. These scaffolds interact with tumor-derived substances; including cytokines, growth factors, and extracellular vesicles, providing dual-action benefits by stimulating osteogenesis and delivering targeted anti-cancer medicines. Scaffolds are designed to release therapeutic compounds in response to certain stimuli, so improving bone regeneration and simultaneously preventing tumor progression by altering cell behavior, angiogenesis, and immune response. Similarly, 3D-bioprinted scaffolds infused with bone marrow-derived mesenchymal stem cells (BMSCs) have shown promising impact in preclinical models for promoting bone regeneration while inhibiting tumor growth [16,17].

Innovative techniques, such as combining BTE constructs with targeted drug delivery, have shown effectiveness in osteosarcoma by achieving localized tumor ablation and bone healing. Furthermore, combinatory techniques, such as photothermal therapy combined with BTE constructs, have demonstrated efficacy in osteosarcoma animals by achieving localized tumor ablation while encouraging bone healing. These innovations underscore the need to design multifunctional BTE systems that address the dual objectives of tumor suppression and bone regeneration [18]. Figure 2 indicates the linkage of BTE with tumor pathologies and interactions.

Additionally, gene therapy has gained immense attention in bone repair and regeneration. It overcomes the limitation of protein delivery and is considered a promising method for maintaining the effective therapeutic concentration of local growth factors in bone defects. The integration of gene therapy with TE termed gene-activated matrix (GAM) scaffolds are utilized for bone regeneration, promoting tissue repair by uniformly incorporating cells within them. GAM may be accessible in many designs depending on the type of material, encoded protein, vector, and way of linking the material and vector [19].

GAM scaffolds promote bone regeneration by incorporating cells. Future studies should explore more advanced biomaterials. The fabrication of GAM with high transfection and long-term gene delivery efficiency with low cytotoxicity is still a challenging job [20].

## 2. Biomaterials

The unmet need for BTE makes it the second most transplanted organ after blood worldwide [21]. Autografts, allografts, and xenografts were the first therapeutic approaches used in bone tissue replacement [22,23]. The market for biomaterials-based orthopedic treatments is rapidly expanding due to the pressing clinical need. Previously, materials for implantation were designed to be ‘bio-inert’. However, materials scientists are now designing ‘bioactive’ materials that integrate with biological molecules or cells and regenerate tissues [24].

In the first Consensus Conference of the European Society for Biomaterials (ESB) in 1976, a biomaterial was defined as a nonviable material used in a medical device, intended to interact with biological systems. Several types of biomaterials (e.g., ceramics, synthetic and natural polymers) are used in the production of scaffolds for TE. Each biomaterial group has specific advantages and, needless to say, disadvantages. The use of composite scaffolds comprised of various phases is becoming increasingly prevalent [25].

### 2.1. Natural Polymers

Natural polymers were the first biodegradable biomaterials used clinically due to their similarities with bone extracellular matrix (ECM) [26]. Compared to conventional synthetic materials, natural polymers exhibit superior chemical versatility, excellent biodegradability, and improved biological performance (i.e., no toxicity, no immunological reactions, and intrinsic osteogenic capacities), which favor cell attachment and promote chemotactic responses [27,28]. The most commonly used natural polymers for building materials for BTE are collagen, gelatin, silk, alginate, cellulose, chitosan, and starch [29]. The natural polymers that are generally used for orthopedic applications are highlighted in Table 1.

### 2.2. Synthetic Polymers

While natural biomaterials offer a favorable physiological context and enhance biological signaling mechanisms in 3D-cultured cancer cells, they each possess inherent limitations. These limitations can include batch-to-batch variability, relatively low elastic modulus (particularly in collagen and Matrigel), and inconsistencies in the presentation of bioactive ligands within 3D matrices. To overcome these limitations, synthetic biopolymers have been developed as a substitute for natural materials. These polymers include poly (ethylene glycol) (PEG), poly (ε-caprolactone) (PCL), poly (vinyl alcohol) (PVA), poly (glycolic acid) (PGA), poly (lactic acid) (PLA), and their derivatives. In addition to PEG-based hydrogels, synthetic materials including PGA, PLA, PVA, and PCL have been used in cancer cell studies. The PGA or PLGA scaffolds could be engineered to have precise degradation rates and mechanical properties, making them suitable for bone regeneration. PLA is extensively used for 3D scaffolds and drug delivery systems. It supports cell attachment and proliferation. Due to ideal elastic properties, PCL is used for scaffolds and as a matrix for the delivery of growth factors and stem cells, respectively. In addition, polymethyl methacrylate (PMMA) is frequently used in bone cement for stabilizing orthopedic implants and as filler in bone defects. Polyurethane (PU) scaffolds can mimic the elasticity of natural bone and have been explored for applications in both soft and hard TE. The silk fibroin could be beneficial in combination with other materials to create scaffolds for bone regeneration.

The components of these polymers are derived from natural metabolites; making them biodegradable and biocompatible in vitro, while also providing essential biological cues to encapsulated cells. Also, PGA has a higher rate of degradation compared to PLA; hence, PGA is used as a co-monomer with PLA to form PLGA to control the rate of degradation of the fabricated scaffolds [41].

### 2.3. Bioactive Ceramics

Bioactive ceramics bond directly with living tissue when implanted. Therefore, they have been extensively studied as biomaterials [25]. Bioceramics can stimulate bone growth by activating the BMP, MAPK, and Wnt/β-catenin pathways when combined with other factors. Bioceramics have certain limitations, such as restricted flexibility and tendency to break, along with challenges related to stem cell growth and difficulty of providing suitable supports for various bone shape. Ceramic bone implants offer long-term benefits like biocompatibility, low cost, osteoconductivity, and corrosion resistance. These biomaterials, composed of inorganic and nonmetallic solids, transform into bone-like structures at elevated temperatures. They have high tensile strength and form stable bonds with the host tissues [42]. Bioceramics, both natural and synthetic, are bioactive materials used in bone tissue repair, due to their high biocompatibility, low immunogenicity, and biodegradability; however, their applications are limited.

Bioceramics alone may not produce the best results for 3D-printed bone scaffolds [43]. As a result, polymers are generally preferred as the organic component in BTE, while bioceramics to serve as the inorganic component. Among the bioceramics frequently utilized for this purpose are hydroxyapatite (HA), β-tricalcium phosphate (β-TCP), bioactive glasses (BGs), and calcium silicate (CaSi). HA has been widely used as ceramic material for fabricating bone scaffolds and implants [44]. HA is used in scaffold designs and dental procedures. Polymers like PLA/HA, silk fibrin/HA, HA/TCP, and collagen/HA/PLA are used as the matrix in scaffold fabrication. Calcium phosphate (CaP), the main component of bone, is used in bone restoration applications due to its mechanical properties. PLA, thermoplastic polyester, is a preferred material in 3D printing technology because of its biocompatibility, high mechanical strength, low cost, and compatibility with drug delivery systems. Blending with HA expands its potential applications in medicine. The optimized PLA/HA composite offers bioactive, osteoinductive, and osteoconductive properties, thereby encouraging future studies. Next to HA, β-TCP is a ceramic material with biocompatibility and biodegradation advantages, promoting bone formation due to its osteoconductive properties. Its enhanced solubility makes it effective for bone grafting in dental and orthopedic applications. The sintering efficiency of β-TCP, synthesized through solid-state reactions, thermal conversion, and precipitation is limited by phase transition, low temperature, and presence of pyrophosphate impurities. However, the efficiency of β-TCP sintering is constrained by three factors: (i) the phase transition β → α-TCP, which occurs at 1115–1150 °C and results in a volume increase, which typically leads to crack formation during the phase transition; (ii) this transition occurs at a relatively low temperature, which hinders the achievement of high densities; and (iii) sintering is sub-optimal when pyrophosphate impurities are present (indicated by a Ca/P molar ratio of less than 1.50). BGs, which are non-porous bioceramics made of silicon dioxide, calcium dioxide, sodium oxide, and phosphorus, are used in biomedical fields for bone tissue repair and regeneration [45].

BGs, when used in the ideal ratio of 50 wt.% SiO_2_, 25 wt.% CaO, and 25 wt.% Na_2_O, enhance biocompatibility and are utilized in biomedical fields for bone tissue repair and regeneration. They bond with bone, stimulate osteogenesis, release ions, and possess antimicrobial properties. Composites combining BGs with polymers or ceramics enhance their mechanical properties and biological performance. However, BGs have limitations, especially in load-bearing applications. Despite their applicability in the human body, they face limitations in material processability and production stages [46].

CaSi bioceramics are gaining interest in bone tissue repair biomaterials due to their high mechanical and bio-interactive properties. They can induce cell differentiation and release calcium and silicon ions, stimulating osteogenic and angiogenic differentiation. However, rapid degradation can lead cytotoxicity and influence cell behavior. Hydrothermal treatment has reduced the degradation of CaSi scaffolds. Natural polymers like peptides, polyesters, and polysaccharides have been used to prepare porous scaffolds for TE. Incorporating CaP in CaSi materials enhances their biological properties and apatite-forming ability [45].

TE is a promising strategy for repairing cartilage and bone tissue. Injectable hydrogels are ideal for 3D cell culture scaffolds due to their high water content, similarity to the natural extracellular matrix (ECM), porous framework for cell transplantation and proliferation, minimal invasive properties, and ability [46]. Nanoclay is used to make hydrogels and porous structures, which improves their mechanical strength, biocompatibility, and capacity to mimic the ECM. These nanoclay-composite hydrogels regulate cell processes such as proliferation, differentiation, and migration, allowing tissue regeneration. Hydrogels inherently possess a 3D network structure that mimics the ECM, allowing cells to grow, migrate, and organize like native tissues. This architecture supports cell attachment and proliferation, essential for tissue formation [47]. The mechanical properties of hydrogels can be tailored through modifications in polymer composition and cross-linking density. For instance, composites that incorporate materials like hydroxyapatite or eggshell particles enhance mechanical stability, making them suitable for load-bearing applications in bone regeneration. Composite hydrogels containing hydroxyapatite not only provide structural support but also release osteogenic factors that encourage bone formation. Studies have demonstrated that these composites can enhance mineralization by pre-osteoblasts without requiring specialized growth media [48].

### 2.4. Biocompatibility

A significant challenge in BTE is developing suitable biomaterials that are biocompatible, biodegradable, and possess appropriate mechanical properties. Various biomaterials, including ceramics, polymers, and composites, have been investigated look into new material combinations. Chitosan, a naturally occurring biopolymer derived from chitin, has garnered attention in BTE due to its biocompatibility, swelling ability, biodegradability, and support for cell adhesion and proliferation. Magnesium abundantly present in the human body has been utilized in biomaterials for orthopedic implants due to its mechanical properties and ability to promote bone growth. Magnesium plays a role in osteoblast differentiation, cellular adhesion, the inhibition of osteoclast activity, and immunomodulation [45,46,47,48,49].

## 3. Scaffold Fabrication Techniques

Tissue-engineered synthetic bone substitutes are artificial scaffolds produced in labs to improve bone remodeling. They act as a 3D bone ECM, supporting cell attachment, mineral deposition, and mechanical support. Stem cells, such as mesenchymal stem cells (MSCs), induced pluripotent stem cells (iPSCs), and embryonic stem cells (ESCs), are important in bone thinning and engineering because of their ability to self-renew and specialize into diverse tissue types. MSCs produced from bone marrow, adipose tissue, or umbilical cord blood are valued for their osteogenic and immunomodulatory capabilities. iPSCs produced by patients can transform into osteoblasts, chondrocytes, or endothelial cells, which help to build bone matrix and vascular structures. ESCs produced from embryos can develop into all germ layers and can produce any cell type necessary for tissue formation [50].

Growth factors, such as vascular endothelial growth factor (VEGF), bone morphogenic proteins (BMPs), insulin-like growth factors (IGFs), and transforming growth factor-beta (TGF-β), and fibroblast growth factors (FGFs) promote cell differentiation and growth. BMPs stimulate bone and cartilage production, whereas VEGF increases angiogenesis, which is essential for scaffold fabrication. BMPs increase osteoinductivity, which encourages MSCs to develop into osteoblasts. IGFs, particularly IGF-1 and IGF-2, play crucial roles in bone formation and development by encouraging the proliferation and differentiation of osteoblast. TGF-β plays a crucial role in cell proliferation, differentiation, and repair, while also modulating immune responses and promoting tissue regeneration. FGFs play important roles in angiogenesis, wound healing, and embryonic development by promoting endothelial cell proliferation and vascularization [51].

### 3.1. Three-Dimensional Printing

Preparing a scaffold requires precise control of material, porous structure, and efficient mass transport. Traditional methods such as solution casting, electrospinning, and lyophilization cannot meet these criteria. Comparison of various 3D printing technologies is illustrated in Table 2. Three-dimensional printing methods have been developed for BTE, providing particularly effective in treating complex bone defects in craniomaxillofacial operations. Additive manufacturing, also known as 3D printing, is a rapid prototype technique that uses computer-aided design (CAD) and computer-aided manufacturing (CAM) technologies to deposit materials layer by layer (LbL) to customize scaffold components. The low incidence of bone cancer, along with need for surgical resection, can lead to delays in diagnosis and treatment. Three-dimensional printing can overcome these limitations by creating scaffolds infused with chemotherapy and growth factors that promote bone regeneration.

Advancements in nanotechnology enable BTE by embedding NPs in 3D-printed scaffolds, enhancing mechanical properties and drug delivery capabilities [51]. Figure 3 indicates the 3D-printed scaffolds infused with NPs for targeted therapeutic action.

### 3.2. Electrospinning

Electrospinning is favored due to its versatility in fabricating randomly oriented fibers. Electrospinning is used to draw micron- and nanometer-sized nonwoven fibers through the electrostatic interactions of charged polymers. The scaffolds created often incorporate hydroxyapatite, a naturally occurring mineral in bone, to enhance osteoconductivity. The process uses an electric charge to draw polymer fibers from a solution, creating a fibrous mat that mimics the extracellular matrix for tissue growth [54].

### 3.3. Phase Separation

A technique that involves the separation of a polymer solution into two phases, leading to the formation of a porous scaffold as the solvent is removed. The nonsolvent-induced phase separation (NIPS) technique, which is normally used for manufacturing membranes, has proven to be an effective method for fabricating composite scaffolds with tunable porosity [55].

### 3.4. Lyophilization

The method involves freezing a polymer solution, followed by the sublimation of the frozen solvent, which results in a porous scaffold structure.

### 3.5. Calcium Phosphate Scaffold Fabrication

The application of 3D scaffolds based on calcium phosphate is extremely promising for BTE as a scaffold material. Calcium phosphate has several advantages as a bone graft material, including biodegradation, biocompatibility, and osteoconductivity. Most synthetic calcium phosphate materials used in clinical practice are based on tricalcium phosphate ceramics, which are reliable and osteoconductive [56].

### 3.6. Polymer Blending

Polymer-based composites are a group of biomaterials that are useful in BTE. Combining different polymers can achieve the desired mechanical and biological properties in the resulting scaffold.

### 3.7. Functionalization of Scaffolds

The scaffold design can be improved by including stem cells that can develop into bone-forming cells, such as adult MSCs, iPSCs, and ESCs. Careful manufacturing and adequate supply of nutrients are necessary. High water-content materials, such as hydrogels, facilitate cell inclusion. For example, Heo et al. successfully encapsulated MSCs and human umbilical vein endothelial cells (HUVECs) in a collagen/fibrin hydrogel, resulting in the MSCs’ development into osteogenic cells [57]. Scaffold-induced cell homing enhances the previous approach by incorporating chemokines and bioactive molecules, attracting stem cells to injury sites using biodegradable scaffolds are strategically placed. Critical molecules in this process include various mimetic peptide sequences, such as RGD (Arg-Gly-Asp), GFOGER (Gly-Phe-Hyp-Gly-Glu-Arg), YIGSR (Tyr-Ile-Gly-Ser-Arg), IKVAV (Ile-Lys-Val-Ala-Val), and REDV (Arg-Glu-Asp-Val). These peptides mimic natural ECM components, providing essential biochemical cues that promote cell attachment to improve osteoblast functionality and overall osteointegration within the host tissue.

Growth factors like BMP, TGF-β, VEGF, and FGF are often used in scaffolds to enhance their osteoinductive and osteoconductive properties. Particularly, BMP and VEGF regulate osteogenesis and angiogenesis, with optimal release rates of VEGF and BMP-2. Specifically, BMP-2 and VEGF were bound to the inner and outer polydopamine (PDA) layers, respectively, which resulted in their sequential adsorption and osteogenic and angiogenic synergy. Various BTE software is illustrated in Table 3. Recent advances in 3D bioprinting combine bioactive ceramics for improved mechanical characteristics and employ biomaterials for scaffolds that enable cell growth and vascularization.

## 4. Advances in BTE

Professor Charles Hull pioneered 3D printing in the 1980s, inventing the first 3D printer [68,69]. Three-dimensional printing technology, also known as additive manufacturing, uses computer-aided design (CAD) software to build 3D models, which are then utilized to manufacture functioning tissue structures with complicated geometries to repair or regenerate damaged tissues and organs [70,71,72]. This technology provides greater precision and reproducibility, allowing for faster material preparation, shorter manufacturing times, and reduced manufacturing costs compared to traditional methods [73,74,75,76]. Tibbits of the Massachusetts Institute of Technology (MIT) demonstrated four-dimensional printing (4D printing) technology at Technology, Entertainment, Design (TED) 2013, allowing printed rope to fold into a three-dimensional structure when immersed in water, clearing the path for more research into 4D printing [77]. Four-dimensional printing enables 3D-printed objects to alter spontaneously and programmatically in response to certain external stimuli, allowing for pre-programmed changes in shape, nature, and scale [78]. Four-dimensional printing enables the preparation of complicated structures that are dynamic and intelligent, thereby accelerating bone healing. Five-dimensional printing employs a five-axis printing process, resulting in multi-dimensional objects that possess superior mechanical qualities while using fewer raw materials than 3D-printed objects. In recent years, 6D printing has emerged as a promising alternative to additive manufacturing technologies with numerous benefits. Additive manufacturing technology offers a wide range of applications [79].

Vascularization plays a crucial role in tissue repair and homeostasis. Neovascularization and blood vessel development are crucial for human growth and tissue repair. The processes of resorption and rebuilding maintain a dynamic balance throughout life. Annually, 25% of bone trabeculae and 3% of cortical bone are replaced, enhancing strength and reducing the risk of fracture through a strong vascular network [80,81,82]. Vascular insufficiency impairs tissue engineering graft survival, demanding early and appropriate vascularization to ensure cell survival and implant integration. Autologous bone grafts are the best bone grafting material for vascularization and osseointegration. The SVVYGLR peptide is an important component of osteopontin (OPN) and has a low molecular weight. This peptide is intended to promote neovascularization and improve internal ischemia in the bone healing material. Unlike RGD, which has a generic pro-adhesive effect, SVVYGLR is a specific pro-vascular peptide. While it did not increase EC proliferation, it significantly improved adhesion and lumen formation activities. Its vascularization effect is comparable to, or perhaps stronger than, that of VEGF. The OPN-derived SVVYGLR peptide, which has both provascularization and osteogenic effects, was covalently bonded to the surface of mesoporous calcium silicate (MCS) powder. This activated the scaffold material, resulting in the creation of an SVVYGLR-MCS composite scaffold using 3D printing technology.

Kuss et al. generated a short-term hypoxic preconditioning approach for 3D-printed scaffolds that uses a bio-ink composed of polycaprolactone/hydroxyapatite and stromal vascular fraction cell-laden hydrogel [83]. The scaffolds were transplanted into athymic mice, indicating the feasibility of prevascularization for 3D bone tissue engineering applications. Chen et al. created PDACS/PCL scaffolds by loading hydrogels with MSCs and ECs [84]. Kan et al. created a 3D-printed bone construct that incorporates an integrated tissue-organ printer [85]. Rukavina et al. used extrusion-based and drop-on-demand bioprinting methods to create pre-vascularized bone tissue using bioink-containing cells [86].

Genetic engineering and regenerative medicine can improve the functionality of vascularized scaffolds. Lin et al. created bone marrow mesenchymal stem cells expressing the BMP-2 gene, which were subsequently encased in 3D-printed hydrogel scaffolds [87]. Cunnife et al. created a gene-activated bio-ink using plasmids, alginate, BMSCs, and nano-hydroxyapatite to promote vascularization and mineralization [88].

Bionics and biomaterials are closely linked in bone tissue engineering, offering better biocompatibility and osseointegration [89]. Bionic scaffolds were developed as a solution to the challenge of achieving centralized vascularization in bone tissue engineering scaffolds. Bionics is classified into three major categories: 3D-printed structures include scaffolds with hollow pipes (“core”), stents with vascular tips (“shell”), and scaffolds with hydrogels (“core”).

Four-dimensional printing technology is an additive manufacturing technology that involves intelligent materials. Intelligent materials that response to external stimuli, are widely used in sensors, actuators, soft robots, medical devices, and artificial muscles, including shape memory polymers (SMPs), shape memory alloys (SMAs), and electroactive polymers (EAPs). Electroactive polymers (EAPs) are a new type of intelligent material having advantages such as being a lightweight, high-performance, and seismically efficient material along with unique electrical and mechanical properties, making them a potential biomimetic material for artificial muscle actuators [90]. Devillard et al. utilized 4D printing to embed thrombin and alkaline phosphatase in bio-ink, enabling localized and pre-programmed calcifications of 3D object parts. This method allows formation of vascularized alveolar bone constructs, offering new directions for multiactive 4D-printed vascularized bone tissue [91]. The research demonstrates that encapsulated alkaline phosphatase can calcify 3D object parts and create fibrin biofilms, enabling the creation of vascularized alveolar bone constructs. This advancement in 4D printing technology could enhance the shape and function of traditional 3D-printed bone structures for future clinical applications. This advancement aligns with the ongoing shift toward personalized and precision medicine, where treatments are increasingly tailored to the needs of individual patients. FDA-approved vascularized 3D-printed bone repair materials are cutting-edge solutions in BTE, designed to mimic natural bone structure and promote vascularization essential for healing. Made from biocompatible materials like hydroxyapatite, bioresorbable polymers, or composites, these scaffolds are precisely customized using advanced 3D printing to fit patient-specific defects. Table 4 illustrates the FDA-approved vascularized 3D-printed bone repair materials.

Five-dimensional and 6D printing technologies are emerging in the revolution of printing technology, while 3D and 4D printing face challenges [92]. Professor William Yerazunis initiated 5D printing, which can be carried out on traditional axes as well as two additional axes, involving the motion of the printing head and the rotation of the print bed [93]. Five-dimensional printing is a new additive manufacturing technology that creates multidimensional products, particularly curved layers, using five-axis printers. It offers a competitive advantage over 3D printing, and can produce stronger objects using less material. Five-dimensional printing is particularly useful for artificial bone production, as it can create complex and strong implants with curved surfaces. Six-dimensional printing, a combination of 4D and 5D printing, offers the ability to print in multiple directions and along complex paths, resulting in more elaborate products that possess both structural integrity and intelligence. It also reduces the use of raw material and provides shorter processing times [94,95]. Five-dimensional and 6D printing have advantages; however, their industrialization is limited due to increased costs, a lack of value for technical progress, and the necessity for precise software and hardware. Furthermore, specialized human resources are required for operation and maintenance, leading to greater time and financial expense. To conquer certain limitations and improve 3D printing in BTE, artificial intelligence (AI), including machine learning (ML), could provide a promising way for materials development and optimization. AI algorithms can analyze vast datasets to identify optimal scaffold designs that enhance mechanical properties and biological compatibility. Techniques such as ML and neural networks could predict how different design parameters affect cell behavior and tissue integration. AI facilitates the creation of customized scaffold geometries based on individual patient anatomy.

**Table 4 ijms-26-03937-t004:** FDA-approved vascularized 3D-printed bone repair materials.

Modification	Properties	Therapy Model	Reference
**Functionalized 3D-printed scaffolds**			
PCL/HA/β-TCP	Biocompatibility, osteoconductivity, controlled porosity, compressive strength (4–8 MPa)	Used for critical-sized bone defects in vivo, promotes osteogenesis	[96]
GelMA/nHA	Hydrophilicity, high cell adhesion, and proliferation, enhanced mechanical properties.	Applied in cranial defect repair in rats; promotes mineralized bone tissue	[97]
PCL-based composites	Enhanced mechanical properties, slow degradation, tailored porosity, FDA-approved	Used in load-bearing bone repair; applied in over 20,000 patients	[98]
PEG-DA/PLGA/nHA	Excellent biodegradability and mechanical support (compressive modulus ~12 MPa)	Tested for bone regeneration under in vitro conditions; osteoinductive	[96]
PCL/Graphene/HA	Superior mechanical strength, hydrophilicity, enhanced electrical conductivity	Promotes bone tissue formation in large defect models	[98]
Ceramic/polymer composites	High compressive strength (~77 MPa for PCL/nHA composites), biodegradability	Applied in orthopedic surgery for repairing fractures and defects	[98]
PLA scaffold/gelatin and Polylysine/BMP-2/VEGF	Sequential release of BMP-2/VEGF in spatiotemporal successfully induced angiogenesis and osteogenesis	In vitro cell experiments	[99]
**Special carrier-loaded 3D-printed scaffolds**			
GelMA Hydrogel-Impregnated PCL Scaffolds	Sustained release of VEGF promotes osteogenesis and angiogenesis	In vitro cell experiments	[100]
β-TCP scaffold/Gel microspheres/Lipo some DFO	Controlled release of DFO promotes osteogenesis and angiogenesis	Rat femoral defect	[100]
PCL scaffold/exosomes/VEGF	Delivery and protection of VEGF promotes osteogenesis and angiogenesis	Rat radial defect	[101]
**Bionic 3D-printed scaffolds**			
β-TCP scaffold/MSCs/ECFCs Hydrogel	Realized central vascularization and Osteogenesis	Rabbit femoral defect	[102]
OCP/GelMA hydrogel/HUVECs scaffold	Simulated bone structure; accelerated osteogenesis and angiogenesis	In vitro cell experiments	[103]
CDHA/axial vascular pedicle scaffold	Simulated bone structure achieved osteogenesis and angiogenesis	Sheep large bone defect	[104]
PLGA/β-TCP/CMs AV bundle scaffold	Combined an AV bundle and rhBMP-2	Rabbit intramuscular pocket	[105]
AKT hollow-channel scaffold	Multi-channel structure achieved osteogenesis and angiogenesis	Rabbit cranial defect; rat muscle implantation	[106]
AKT/bio-ceramic/bioactive glass scaffold	Haversian bone-mimicking scaffold promoted osteogenesis and angiogenesis	Rabbit femoral defect	[107]

Traditional therapies that combine surgery, chemotherapy, and radiation have reached their limits of efficacy, motivating efforts to develop new therapeutic methods. The development of biomaterials offers innovative options for the treatment of bone tumors. Suitable biomaterials can simultaneously provide tumor therapy and promote bone regeneration.

Recent advances in BTE for cancer patients incorporate NMs, smart materials, 3D bioprinting, electrospinning, and drug delivery systems. Additionally, AI, machine learning, and CRISPR-based gene editing enhance scaffold design, personalize treatments, and modify stem cell activity for the management of oncological patients [108,109,110,111,112,113]. The integration of advanced biomaterials, innovative techniques, and bioactive molecules in BTE presents a promising avenue for addressing the unique challenges faced by oncological patients. Biomaterials, techniques, and molecules that link the implications of BTE in oncological patients are depicted in Table 5.

## 5. Conclusions and Future Prospects

In conclusion, it will be vital to address the multifaceted challenges posed by bone tumors and to enhance therapeutic strategies through pioneering research in TE, biomaterials, and advanced manufacturing technologies as these efforts will shape the future landscape of cancer treatment. The development of multi-functional, biocompatible scaffolds has become an essential focus in the field of BTE. The potential of 3D and 4D printing technologies to create personalized scaffolds tailored to individual anatomical and biological needs promises to revolutionize traditional treatment paradigms in orthopedic oncology. As these innovations continue to evolve, they promise to significantly improve the quality of life and survival rates for patients with bone cancers. Moreover, the role of AI and machine learning in BTE is expected to expand, enabling the optimization of scaffold design and material selection through the analysis of extensive datasets. AI applications may facilitate rapid prototyping and iterative design processes, ultimately leading to in faster translations of innovations from the lab to the clinic.

Future studies should prioritize investigating new biomaterials that not only possess biocompatible and osteoconductive properties but also have bioactive reinforcement through gene editing and enhanced growth factor delivery systems. Investigating patient-specific solutions using advanced imaging technologies and adaptive manufacturing strategies, such as 6D printing, may further enhance the dynamic capabilities of the field.

## Figures and Tables

**Figure 1 ijms-26-03937-f001:**
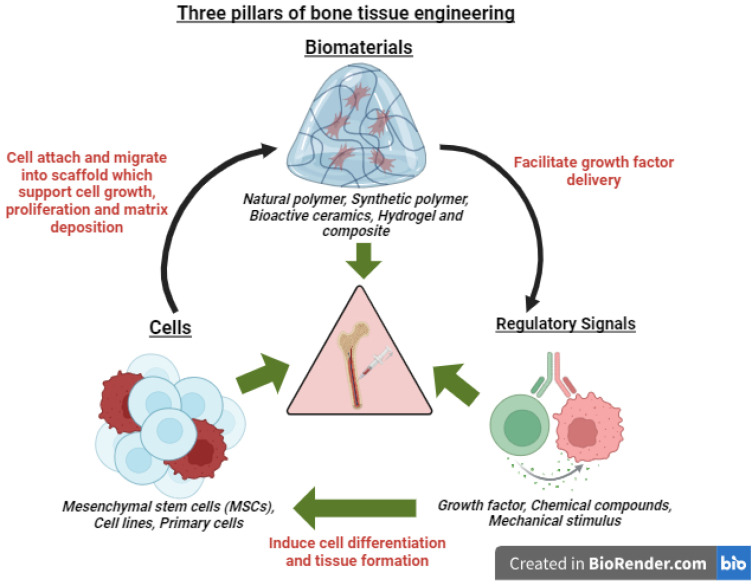
The three essential pillars supporting bone regeneration.

**Figure 2 ijms-26-03937-f002:**
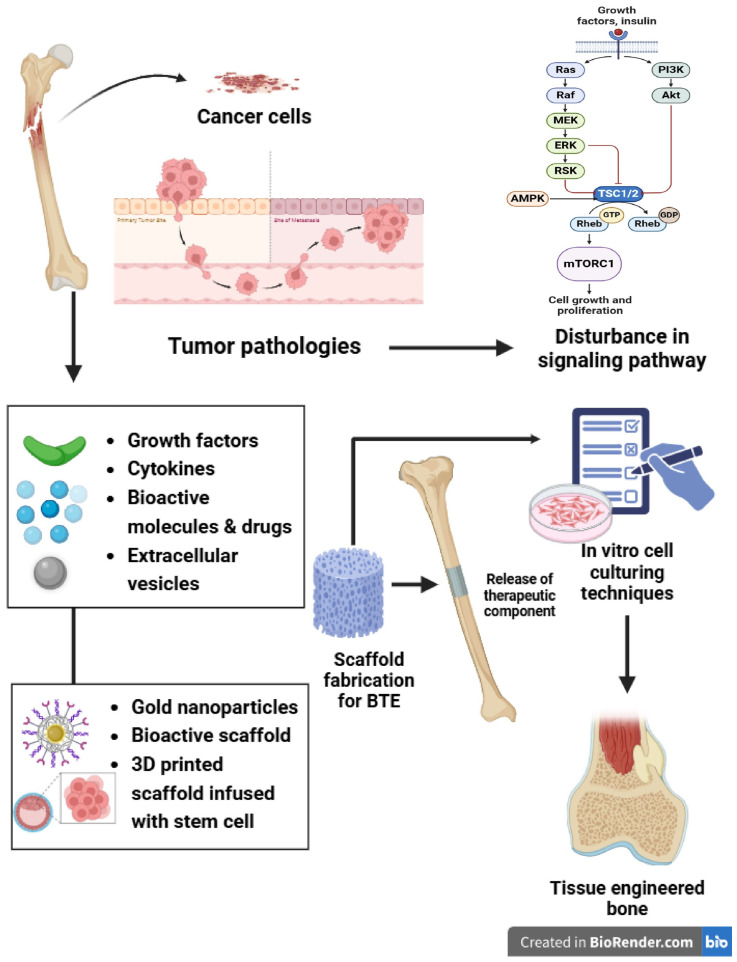
Linkage of bone tissue engineering with tumor pathologies and interactions.

**Figure 3 ijms-26-03937-f003:**
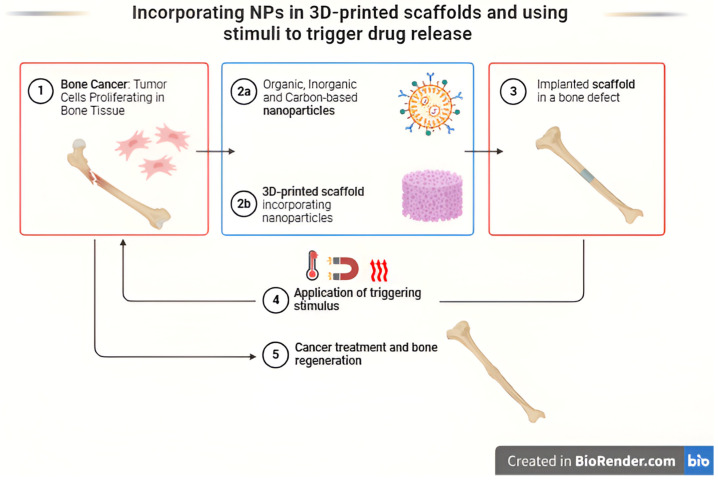
Three-dimensionally printed scaffolds infused with NPs for targeted therapeutic action [51].

**Table 1 ijms-26-03937-t001:** Natural polymers for orthopedic application.

Polymer	Source	Advantages	Disadvantages	References
Natural polymer				
Collagen	Ovine, porcine, equine, and bovine	Biocompatibility promotes cell adhesion and growth, biodegradable.	Relatively weak mechanical stiffness, risk of immunogenic reactions, potential disease transmission.	[30]
Gelatin	Denaturation and hydrolysis of natural collagen	Biocompatible, biodegradable, cell-binding properties, low cost, gelling properties	Low mechanical properties and fast degradation rate.	[31]
Silk	Arthropods (Lepidoptera larvae such as silkworms; arachnids such as spiders, mites, and some scorpions; and some flies)	Biocompatible, biodegradable, flexible processability, high mechanical strength, thermally stable	Slow degradation rate (2–4 years)	[32,33,34]
Alginate	Brown seaweed	Biocompatible, biodegradable, relatively low cost, easy gelatin by ionic cross-linking, and easy chemical modification with adhesion via RGD ligands	Poor cell-material interaction due to inherent lack of cell adhesivity and low protein adsorption	[35]
Cellulose	Wood, plants, tunicates, and algae	Biodegradable, biocompatible, high mechanical performance	Low cell-binding properties	[36,37]
Chitosan	Exoskeleton of crustaceans and mollusks, insect cuticles, and fungi	Bioactive, biocompatible, biodegradable, antibacterial, and nonimmunogenic properties; the ability for cell ingrowth	Relatively weak mechanical strength and stability	[38,39]
Starch	Corn, potato, wheat, and tapioca	Biocompatible, biodegradable, low cost	Lack of processability, low surface area	[27,40]

**Table 2 ijms-26-03937-t002:** Comparison of various 3D printing technologies.

Technique	Materials Used	Working Principle	Advantages	Disadvantages	Schematic
Fused deposition modeling (FDM)/Fused filament fabrication (FFF) [51]	Acrylonitrile butadiene styrene (ABS), PLA, PCL, polyethylene terephthalate glycol (PET-G), tricalcium phosphate (TCP), nylon	FDM works by extruding a thermoplastic or composite filament layer by layer to create customized, porous scaffolds for bone tissue engineering, enabling cell infiltration, nutrient diffusion, and osteogenesis.	Customizable scaffold design’ Material versatility’ FDM is relatively affordable, easy to use, and scalable, making it suitable for producing prototypes and large quantities of bone scaffolds efficiently.	FDM-printed scaffolds may have lower mechanical properties compared to native bone; The high temperatures used can denature bioactive molecules, limiting their direct inclusion during fabrication.	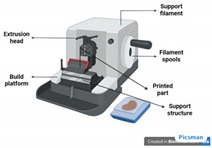
Pressure-assisted microsyringes (PAMs) [52]	A semi-liquid mixture of polymers and solvents (solution, paste, or dispersion)	Pressure-assisted microsyringes (PAMs) work by extruding bio-inks or biomaterials through a microsyringe nozzle under controlled pressure to fabricate 3D scaffolds layer by layer.	Enhanced biocompatibility; Operates at room temperature; Suitable for bioprinting; Continuous flow of aqueous-based materials.	Quality depends on rheological properties; Requires solvents or crosslinking agents, slower printing speed.	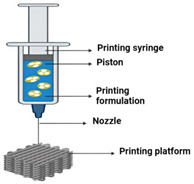
Stereolithography (SLA) [53]	Photo-curable liquid resin	Uses light sources from UV to visible light to crosslink or polymerize the ink for the development of 3D structures.	High resolution and precision; Biocompatible photopolymers and hydrogels are used; Precise control over pore size and distribution for better tissue integration and regeneration outcomes.	Limited material options for biodegradability; Post-processing requirements costly compared to other scaffold fabrication methods.	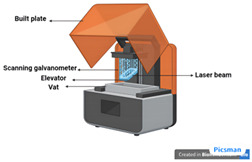
Laser-assisted method [51]	Ink solution, laser energy-absorbing powders	Based on laser-induced forward transfer (LIFT) effect. A NIR or UV pulsed laser is used that transfers energy into a liquid photopolymerizable material. Photopolymerization occurs, and the product is created LbL.	Laser techniques can be controlled to provide minimal heat transfer to surrounding tissues; High resolution; Solvent-free process.	High-energy lasers might degrade drugs; Only laser energy-absorbing components can be used; Expensive setup.	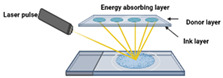
Binder jetting [52]	Binder fluid, powder bed	Binder jetting involves depositing a liquid binder onto a powder material, layer by layer, to create scaffolds. The powder typically includes bioactive ceramics or polymers, and the binder fuses the particles. Post-processing like sintering enhances the scaffold’s mechanical properties for bone regeneration.	Can be performed at room temperature; Wide range of starting materials; Fast-disintegrating dosage forms can be produced.	Post-fabrication processes necessary; Use of organic solvents; Wastage of powder material; Results in fragile dosage forms.	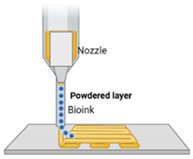
Inkjet printing [51]	Ink–drug solution, substrate–polymer-based films	Two-step process: (1) formation of electrostatically charged ink droplets and directing them toward predefined locations on the substrate and (2) droplet and substrate get to interact.	Precision in layer by layer fabrication; Scalability and cost-effectiveness.	Potential for nozzle clogging and droplet inconsistencies; Restricted material viscosity range.	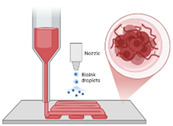

**Table 3 ijms-26-03937-t003:** Software and its applications in bone tissue engineering (BTE).

List of Software	Method of Fabrication	Materials	Application	Reference
SolidWorks	3D-printing	Polycaprolactone (PCL)	Designing load-bearing scaffolds for critical-sized bone defects	[58]
Autodesk Fusion 360	Selective Laser Sintering (SLS)	Poly(L-lactide) (PLLA)	Fabrication of scaffolds for large bone defect repair, particularly in long bones	[59]
MATLAB	Electrospinning	Poly (lactic-co-glycolic acid) (PLGA)	Developing nanofiber structures for bone cell attachment and proliferation	[60]
ANSYS	Bioprinting	Hydrogel + Mesenchymal Stem Cells	Enhancing osteogenesis by creating stem cell-laden scaffolds for bone regeneration	[61]
Rhino + Grasshopper	Stereolithography (SLA)	Hydroxyapatite (HA)	Creating high-resolution scaffolds for craniofacial bone reconstruction	[62]
BioCAD	Direct Ink Writing	Collagen + Bioactive Glass	Printing composite scaffolds for repairing segmental bone defects	[63]
Blender	Fused Deposition Modeling (FDM)	Polyethylene Glycol (PEG)	Producing scaffolds for filling irregular bone defects in orthopedic surgery	[64]
Mimics	Computer-Aided Design (CAD)	Calcium Phosphate (CaP)	Generating patient-specific scaffolds for complex bone reconstruction surgeries.	[65]
COMSOL Multiphysics	Melt Electrospinning	Polycaprolactone (PCL) + Gelatin	Optimizing mechanical properties of scaffolds for vertebral bone repair	[66]
Simpleware	Laser-Assisted Bioprinting	Bioink + Osteoblast cells	Fabricating osteoblast-laden scaffolds for in vitro studies of bone formation	[67]

**Table 5 ijms-26-03937-t005:** Biomaterials, techniques, and molecules that link the implications of BTE in oncological patients.

Biomaterials for BTE	Cutting-Edge Techniques in BTE	Emerging Molecules in BTE	Oncological Patient	Reference
**Natural Biomaterials**				
Collagen	Three-dimensional bioprinting	Bone Morphogenetic Proteins (BMPs)	Patients with bone defects post-tumor resection or bone metastasis (e.g., breast cancer)	[114]
Gelatin	Decellularized scaffolds	Vascular Endothelial Growth Factor (VEGF)	Patients with bone defects caused by metastatic cancers (e.g., lung, prostate)	[115]
Chitosan	Controlled drug Delivery Systems	Mesenchymal Stem Cells (MSCs)	Patients undergoing chemotherapy with compromised immune systems (e.g., lymphoma)	[116]
Alginate	decellularized scaffolds	Gene Therapy	Patients with metastatic bone disease needing bone regeneration (e.g., multiple myeloma)	[117]
Hydroxyapatite	Three-dimensional bioprinting	BMPs and VEGF combined	Patients with large bone defects from tumor excision (e.g., osteosarcoma)	[118]
Bacterial cellulose	Three-dimensional bioprinting	BMP-2	Patients with bone defects post-tumor resection (e.g., osteosarcoma)	[119]
Hyaluronic acid (HA)	Hydrogel formation	HA	Post-operative patients requiring tissue repair (e.g., colorectal cancer)	[120]
Cerium oxide-containing beads	Antioxidant activity enhancement	Cerium Oxides (Ce^3+^/Ce^4+^)	Patients with bone defects and oxidative stress (e.g., breast cancer)	[121]
**Synthetic Biomaterials**				
Poly (lactic-co-glycolic acid) (PLGA)	Cell-aligned HDGs	Bone Morphogenetic Proteins (BMPs): BMP-2 and BMP-7	Patients with bone defects post-tumor resection or bone metastasis (e.g., breast cancer)	[122]
Polycaprolactone (PCL)	Decellularized scaffolds	Vascular Endothelial Growth Factor (VEGF)	Patients with bone defects due to metastases (e.g., prostate, lung cancers)	[123]
Polyethylene glycol (PEG)	Controlled drug delivery systems	Mesenchymal Stem Cells (MSCs)	Patients undergoing chemotherapy with bone regeneration needs (e.g., lymphoma, leukemia)	[124]
Polylactic acid (PLA)	Three-dimensional bioprinting	Gene Therapy	Patients with bone loss from tumor resection or radiation (e.g., head and neck cancers)	[125]
Polymethyl methacrylate (PMMA)	Electrospinning	BMPs and VEGF combined	Patients with critical bone defects after excision of bone tumors (e.g., osteosarcoma)	[126]
Poly-D, L-lactic acid (PDLLA)	Bioactive coatings	Growth Factors (BMP-2, TGF-β)	Patients with post-surgical bone loss due to cancer resection (e.g., bone metastasis)	[127]
Polyurethane (PU)	In situ gelation	Calcium Phosphate Compounds	Patients with bone damage from radiation therapy or metastatic bone disease (e.g., multiple myeloma)	[128]
**Nanomaterials and Smart Materials**				
Nano-hydroxyapatite (nHA)	Electrospinning, nanocomposite formation, AI-based predictive models for scaffold design	BMP-2, VEGF, MSCs	Patients with bone-related disorders like tumors, metastases, and osteoporosis.	[108]
Graphene-based nanomaterials	Three-dimensional bioprinting, nanocoating, machine learning for material optimization	VEGF, BMP-2, TGF-β, IGF-1	Patients undergoing chemotherapy with bone metastasis and osteosarcoma.	[109]
Carbon nanotubes (CNTs)	Functionalization, composite formation, AI and ML algorithms for optimizing CNT loading in scaffolds	BMP-2, VEGF, FGF	Patients with bone defects due to cancer treatment (e.g., osteosarcoma, multiple myeloma)	[110]
Silver nanoparticles (AgNPs)	Nanofabrication, controlled drug release	BMP-2, VEGF, Anticancer Drugs	Patients with bone infections or metastasis (e.g., lung cancer with bone metastases)	[14]
Mesoporous silica nanoparticles (MSNs)	drug delivery systems, surface modification, CRISPR-based gene editing.	BMP-2, TGF-β, IGF-1	Patients with bone regeneration issues after cancer surgery or metastasis (e.g., colorectal cancer)	[14]
Thermoresponsive hydrogels	Thermoresponsive drug delivery, injectable hydrogels	BMP-2, VEGF, MSCs	Patients with bone fractures after tumor excision or chemotherapy (e.g., lymphoma, leukemia)	[111]
Magnetic nanoparticles (MNPs)	Magnetic targeting, magnetically induced hyperthermia, AI-assisted magnetic field optimization	VEGF, BMP-2, Growth Factors	Patients with bone defects post-cancer treatment or bone metastases.	[112]
Polymeric nanogels	Nanogel formation, drug delivery systems, CRISPR/Cas9-based gene editing to modify stem cell behavior for bone regeneration	BMP-2, VEGF, TGF-β, FGF	Patients with critical bone defects after tumor surgery or metastasis (e.g., ovarian cancer)	[113]

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
