# Peer review of "Innovative Approaches in Bone Tissue Engineering: Strategies for Cancer Treatment and Recovery"

_ijms, 2025, doi:10.3390/ijms26093937_

Round 1

Reviewer 1 Report

Comments and Suggestions for Authors

This manuscript discusses the latest advances in BTE, focusing on scaffold fabrication techniques, the application of stem cell and gene therapies, and innovations in biodegradable materials. It also explores the potential of nanotechnology in targeted drug delivery and looks ahead to the use of artificial intelligence in scaffold design to drive the development of personalized therapeutic strategies for bone cancers. This manuscript is rich in content, logically rigorous, and meticulously crafted, but there are still a few minor issues that need to be revised, as outlined below:

1. The captions for the two figures in the manuscript are quite simple. Adding more explanation could enhance the reader's understanding.

2. The introduction is rich in content, but it is too detailed, which may make it difficult for readers to grasp the main focus. It is recommended to make it more concise and to highlight the theme more clearly.

3. The content in the 2nd section is quite limited. Please consider whether it is appropriate to present it as a separate section.

4. In section 3.2, there are likely many synthetic materials as well. Why weren't they discussed in more detail?

5. Should "4.1. D printing" be "4.1. 3D printing"?

6. Section 4 discusses scaffold fabrication techniques but only covers 3D printing. Other fabrication techniques should also be mentioned.

Author Response

This manuscript discusses the latest advances in BTE, focusing on scaffold fabrication techniques, the application of stem cell and gene therapies, and innovations in biodegradable materials. It also explores the potential of nanotechnology in targeted drug delivery and looks ahead to the use of artificial intelligence in scaffold design to drive the development of personalized therapeutic strategies for bone cancers. This manuscript is rich in content, logically rigorous, and meticulously crafted, but there are still a few minor issues that need to be revised, as outlined below:

Overall response: The authors would like to thank the editor’s and reviewers’ critical comments and encouragement. By following their suggestion, we believe the quality of the manuscript could be greatly improved. The revisions are shown in red in the revised manuscript.

Comment 1. The captions for the two figures in the manuscript are quite simple. Adding more explanation could enhance the reader's understanding.

Response: We appreciate the reviewer’s suggestion for improving the quality of the figures. As suggested, the authors have made informative changes to all the figure captions in the revised version of the manuscript.

Comment 2. The introduction is rich in content, but it is too detailed, which may make it difficult for readers to grasp the main focus. It is recommended to make it more concise and to highlight the theme more clearly.

Response: We appreciate the reviewer’s suggestion for betterment of manuscript. As suggested, authors made the necessary changes in the introduction part in the revised version of the manuscript.

Comment 3. The content in the 2nd section is quite limited. Please consider whether it is appropriate to present it as a separate section.

Response: We appreciate the reviewer’s suggestion for betterment of manuscript and avoid further confusions. As suggested, authors made the necessary changes and club both the sections in the revised version of the manuscript.

Comment 4. In section 3.2, there are likely many synthetic materials as well. Why weren't they discussed in more detail?

Response: We appreciate the reviewer’s suggestion for the betterment of manuscript. As suggested, authors have discussed the section 3.2 in detail in the revised version of the manuscript. Authors have added polymethyl methacrylate (PMMA), Polyurethanes (PU), and silk fibroin with their applications in the BTE. Section 3.2 should be changed to section 2.2 in the revised version of the manuscript.

Comment 5. Should "4.1. D printing" be "4.1. 3D printing"?

Response: We appreciate the reviewer’s suggestion for betterment of manuscript. As we apologise for the typological mistakes and corrected in the revised version of the manuscript.

Comment 6.  Section 4 discusses scaffold fabrication techniques but only covers 3D printing. Other fabrication techniques should also be mentioned.

Response: We appreciate the reviewer’s suggestion for the betterment of manuscript. The authors have briefly included other important fabrication techniques in the revised version of the manuscript.

Reviewer 2 Report

Comments and Suggestions for Authors

The review lacks the Purpose. It is not stated in Introduction. Also, Introduction need a short description on what is to be expected in-text.

Subtitle 2. - there is substantial information, but no reference has been included. Also - 116-118, personal beliefs are to be discussed in Discussions.

121- 122 - repetition..

133 - typing mistake?

All manuscript must undergo grammar and spelling revision.

170 - Stem cells, medications, and other resources. - Is that to be considered a subchapter?

4.1. D printing or 3D?

286 - is a personal belief or found in literature?!

Figure 2 - is a reproduction. Is there an approval from the original authors? Same question for the images used in Table 2.

Table 2 must be revised.

The manuscript is consistent regarding bone tissue engineering, but unfortunalety, there are no links with tumoral pathologies effects/interractions/discussions/studies/perspectives. Either the authors would include a chapter to be adressed to, either change the title and submit the manuscript to a specific journal.

Comments on the Quality of English Language

The manuscript should be revised for grammar and spelling.

Author Response

Overall response: The authors would like to thank the editor’s and reviewers’ critical comments and encouragement. By following their suggestion, we believe the quality of the manuscript could be greatly improved. The revisions are shown in red in the revised manuscript.

Comment 1. The review lacks the Purpose. It is not stated in Introduction. Also, Introduction needs a short description on what is to be expected in-text.

Response: Authors appreciate the significant comments from the reviewers which probably improve the quality of the manuscript. Authors have tried to depict the main purpose of the article in the revise version of the manuscript. It explores and develops innovative approaches in bone tissue engineering (BTE) that specifically addresses the unique challenges faced in cancer treatment and recovery.

Comment 2. Subtitle 2. - there is substantial information, but no reference has been included. Also - 116-118, personal beliefs are to be discussed in Discussions.

Response: Authors appreciate the significant comments from the reviewers. Authors have cited the latest references for the same and eliminated the 116-118 lines from the discussion in the revise version of the manuscript.

Comment 3. 121- 122 - repetition..

Response: Authors appreciate the significant comments from the reviewers. Authors have made the suggested corrections and removed the duplicate references in the revise version of the manuscript.

Comment 4. 133 - typing mistake?

Response: Authors appreciate the significant comments from the reviewers. Authors have made the suggested corrections in the revise version of the manuscript.

Comment 5. All manuscript must undergo grammar and spelling revision.

Response: The authors appreciate the significant comments from the reviewers that improve the quality of the manuscript. They have undergone stringent language and grammar corrections by professionals in the revised version of the manuscript. All the changes are highlighted in the red color.

Comment 6. 170 - Stem cells, medications, and other resources. - Is that to be considered a subchapter?

Response: Authors appreciate the significant comments from the reviewers which improve the quality of the manuscript. Authors thought it should be fine to keep manuscript brief. Still reviewers wanted to subsection, in that case authors are bind to the suggestions given by reviewers.

Comment 7. 4.1. D printing or 3D?

Response: Authors appreciate the significant comments from the reviewers. Authors have made the suggested corrections in the revise version of the manuscript.

Comment 8. 286 - is a personal belief or found in literature?!

Response: Authors appreciate the significant comments from the reviewers. Authors have made the suggested corrections and delete the sentence in the revise version of the manuscript.

Comment 9. Figure 2 - is a reproduction. Is there an approval from the original authors? Same question for the images used in Table 2.

Response: Authors appreciate the significant comments from the reviewers and aware about the copyright violations. Figure 2 is drawn with the help of Biorender with plenty of modifications. The images present in Table 2 are drawn with the help of smart software’s and biorender and not subjected to any copyrights.

Comment 10. Table 2 must be revised.

Response: Authors appreciate the significant comments from the reviewers. We have carefully revised Table 2 in the revised version of the manuscript.

Comment 11. The manuscript is consistent regarding bone tissue engineering, but unfortunately, there are no links with tumoral pathologies effects/interractions/discussions/studies/perspectives. Either the authors would include a chapter to be adressed to, either change the title and submit the manuscript to a specific journal.

Response: The authors appreciate the significant comments from the reviewers. Specifically, the authors wanted to draw attention to the applications and future challenges of bone tissue engineering in cancer therapy. The authors agree with the reviewers' concerns and have aimed to address these comments by briefly adding necessary insights regarding the connections between tumoral pathologies and their effects in the revised version of the manuscript (introduction section). Additionally, we have included a figure that illustrates the brief relationship between tumoral pathologies and advance strategies.

Round 2

Reviewer 2 Report

Comments and Suggestions for Authors

The manuscript has been improved after adjustments.

Still, the authors must include in the subsections of the evaluated biomaterials, techniques, molecules, paragraphs which would link the implications of the BTE in oncological patients.

L463 - future prospective does not need capital letters.

Author Response

Overall response: The authors would like to thank the editor’s and reviewers’ critical comments and encouragement. By following their suggestion, we believe the quality of the manuscript could be greatly improved. The revisions are shown in red in the revised manuscript.

Comment 1. The authors must include in the subsections of the evaluated biomaterials, techniques, molecules, paragraphs which would link the implications of the BTE in oncological patients.

Response 1: Authors appreciate the significant comments from the reviewers which probably improve the quality of the manuscript. We already mentioned the biomaterials and their techniques, such as 3D bioprinting, decellularized scaffolds, electrospinning, phase separation, lyophilization, and calcium phosphate scaffold fabrication, in the first revision of the manuscript. In response to the reviewers' comments, we have now added biomaterials, cutting-edge techniques, and molecules that link the implications of bone tissue engineering (BTE) in oncology. Additionally, we have compiled all of this information in Table 4 and 5 in the revised version of the manuscript.

Comment 2.  L463 - future prospective does not need capital letters.

Response 2: Authors appreciate the significant comments from the reviewers. We have made the necessary changes in the revised version of the manuscript.

Round 3

Reviewer 2 Report

Comments and Suggestions for Authors

The manuscript has been improved after revising following comments.